# Reproducibility of "ITI-GEN: Inclusive Text-to-Image Generation"

## Abstract

A major limitation of current text-to-image generation models is their inherent tendency to incorporate biases, thereby not demonstrating inclusivity in certain attributes. An approach to enhance the inclusiveness is suggested by Zhang et al. (2023): Inclusive Text-to-Image Generation (ITI-GEN). The authors state that ITI-GEN leverages reference images to improve the inclusiveness of text-to-image generation by learning inclusive prompt embeddings for targeted attributes. In this paper, the reproducibility of ITI-GEN is investigated in an attempt to validate the main claims presented by the authors. Moreover, additional experiments are conducted to provide further evidence supporting their assertions and research their limitations. This concerns the research on inclusive prompt embeddings, the inclusivity of untargeted attributes and the influence of the reference images. The results from the reproducibility study mainly show support for their claims. The additional experiments reveal that ITI-GEN only guarantees inclusivity for the specified targeted attributes. To address this shortcoming, we present a possible solution, namely ensuring a balanced reference dataset.

## 1 Introduction

Recently, there has been a significant advancement in text-based visual content creation, driven by breakthroughs in generative modelling and the utilisation of large-scale multimodal datasets (Sohl-Dickstein et al., 2015; Ho et al., 2020; Ramesh et al., 2021). These developments have reached a point where publicly released models, such as Stable Diffusion (SD) (Rombach et al., 2022), can now generate highly realistic images based on human-written prompts. However, a major limitation of current models lies in their inherent tendency to incorporate biases from the training data, thereby not yet demonstrating inclusivity in certain attributes and minority groups (Bianchi et al., 2023; Ramesh et al., 2022; Bansal et al., 2022).

A straightforward approach to enhance inclusivity in text-to-image generation is the usage of a truly inclusive training dataset (Dhariwal & Nichol, 2021; Zhao et al., 2020). However, obtaining an extensive training dataset that is well-balanced across all relevant attributes is deemed impractical. An alternative strategy to achieve more inclusiveness is to specify or enumerate each category in natural language, i.e. hard prompt searching (HPS) (Bansal et al., 2022; Hutchinson et al., 2022; Ding et al., 2021; Petsiuk et al., 2022). Nevertheless, numerous categories are difficult to specify with natural language, such as skin tone, or cannot be well synthesised by the existing models due to linguistic ambiguity or model misrepresentation.

Contrary to these two approaches, Zhang et al. (2023) suggest a more efficient strategy called Inclusive Text-to-Image Generation (ITI-GEN). This method utilises images as guides to enhance representation by learning inclusive prompt embeddings. The authors claim that ITI-GEN enhances the inclusiveness by generating images with a uniformly distributed representation across attributes of interest.

In this work, we aim to reproduce their findings, verify their claims, and conduct additional experiments to provide further evidence supporting their assertions and research their limitations.

## 2 Scope of reproducibility

Zhang et al. (2023) propose a novel method for inclusive text-to-image generation, in which inclusive prompt embeddings are learned using images. With this method they achieve group fairness (Mehrabi et al., 2021) for the specified groups in the inclusive prompts. For the purpose of our discussion, we denote groups as attributes (e.g. gender, age), encompassing categories (e.g. male, 10-19). To maintain consistency with the terminology used in the original paper, we consider the following scenarios: a *single binary attribute* pertains to one attribute that includes two categories, *multiple attributes* involve a combination of binary attributes, and *multi-category attributes* encompass a combination of attributes with multiple categories.

The main claims of the original paper that we address can be summarised as follows:

- **Claim 1: Inclusiveness in text-to-image generation:** ITI-GEN can generate images that represent a more uniform distribution across different binary attributes, improving inclusiveness in text-to-image generation compared to state-of-the-art approaches that enumerate each category in natural language.

- **Claim 2: Scalable to multiple attributes:** ITI-GEN can generate a more evenly distributed representation across various category combinations of binary attributes compared to baseline methods.

- **Claim 3: Scalable to multi-category attributes:** ITI-GEN achieves more inclusiveness compared to baseline methods for attributes encompassing multiple categories, including those that are notably underrepresented in the training data.

- **Claim 4: High-quality images:** ITI-GEN generates images with FID scores of approximately 60, indicating high-quality images.

In addition to reproducing the results presented in the paper, we perform novel experiments. We investigate the disparities between the embeddings produced by ITI-GEN and those derived from natural language. Additionally, we delve into a limitation of ITI-GEN: the possibility of biases being introduced by the reference images, as indicated by Zhang et al. (2023).

## 3 Methodology

The ITI-GEN implementation is publicly accessible on GitHub within their repository[1]. The replication of all experiments is facilitated by the provided code, slight modifications were introduced to ensure compatibility with our GPU server infrastructure[2]. Additionally, we conducted experiments extending beyond the scope of their original study to gain further insights into the performance of ITI-GEN.

### 3.1 Model descriptions

The original paper proposes ITI-GEN, a new framework that achieves inclusiveness in text-to-image generation by creating discriminative prompts based on reference images. A general overview of image generation using ITI-GEN prompts is provided in Appendix A. To realise inclusiveness, ITI-GEN aims to sample an equal number of images capable of representing any category combination within the attribute set $A$. Here, $A$ encompasses $M$ distinct attributes (e.g. gender, age), each containing $K_m$ categories (e.g. types of gender). This process involves utilising a pre-trained generative model, namely SD (Rombach et al., 2022) (sd-v1-4), and a human-written prompt tokenised as $\mathbf{T} \in \mathbb{R}^{p \times e}$ obtained with the vision-language pre-trained CLIP model (Radford et al., 2021). SD and CLIP have a total of 860M and 428M parameters, respectively.

For each mutually exclusive category $k$, within attribute $m$, inclusive prompt embeddings are acquired. This is done by introducing $q$ learnable tokens $\mathbf{S}_k^m \in \mathbb{R}^{q \times e}$ after the original token $\mathbf{T}$, thereby constructing a new

---

[1] https://github.com/humansensinglab/ITI-GEN
[2] Project page: https://anonymous.4open.science/r/factai-2422

prompt $\mathbf{P}_k^m = [\mathbf{T}; \mathbf{S}_k^m]$. The default CLIP embedding dimension $e$ is 768 and the authors set $q$ to 3, making the total number of parameters $\sum_{m=1}^M K_m \times 3 \times 768$.

In the presence of a reference image set corresponding to target attributes, the learnable prompts undergo a training process designed to align these attributes within the images. This alignment is achieved through the application of both Direction Alignment Loss and Semantic Consistency Loss. The former ensures the alignment of directions between the prompts $\mathbf{P}_i^m$ and $\mathbf{P}_j^m$ with the direction defined by the embeddings of the reference images. The latter focuses on maximising the cosine similarity between the learning prompts and the original input prompt. This serves the purpose of maintaining faithfulness to the input prompt during the generation of images.

### 3.2 Datasets

The original paper conducted experiments using a total of four datasets, whereas this research exclusively utilises two of these datasets for training the inclusive prompt embeddings. (1) CelebA contains images featuring faces of celebrities annotated with 40 binary attributes, such as *male*, *young* and *eyeglasses* (Liu et al., 2015). In total, CelebA consists of 16,000 images, with 200 images per category. (2) FairFace includes images of faces annotated with two perceived gender categories and nine perceived age categories (Karkkainen & Joo, 2021). This dataset contains a total of 2,200 images, with 200 images per category. The authors have made these datasets available in a dedicated repository[3].

### 3.3 Evaluation

The generated images are evaluated with the following two metrics. (1) *Distribution Discrepancy* ($\mathbb{D}_{\mathrm{KL}}$) is used to quantify the distribution diversity of the attributes. The original paper aims to generate inclusive images, ensuring a uniform distribution across the attributes of interest. To classify the attributes in the images two models are used, namely CLIP for the binary attributes and a classifier proposed by Karkkainen & Joo (2021) for the multi-category attributes. (2) *Fréchet Inception Distance* (FID) is used to measure image quality (Heusel et al., 2017).

### 3.4 Hyperparameters

In order to closely replicate the original experiments, we predominantly employ the same set of hyperparameters. Due to limited time and resources, the Denoising Diffusion Implicit Models (DDIM) steps of SD is decreased from 50 to 25 for the experiment of the third claim. The DDIM framework iteratively improves samples via denoising. Reducing its steps from 50 to 25 speeds up the sampling process but could slightly alter the quality. However, SD would still produce satisfactory images (Song et al., 2020).

### 3.5 Experimental setup and code

**Experiments - Claim 1 & 2** This experiment researches the inclusiveness of ITI-GEN and its scalability to multiple attributes. Due to limitations in GPU resources, we were unable to execute every experiment with each of the five prompts outlined in the original paper. Consequently, we conducted the experiments exclusively using the prompt "a headshot of a person". As a result, we established our own baseline for Vanilla SD and HPS.

we trained the ITI-GEN model, following the methodology and code from the original experiments. For each attribute, we generated 200 images, categorising them into positive and negative instances. Subsequently, we conducted a comparative analysis between ITI-GEN, SD and HPS baselines.

**Experiment - Claim 3** This experiment investigates the scalability of ITI-GEN to multi-category attributes. Specifically, we reproduced one of the two challenging settings proposed in the original paper, namely Perceived *Gender × Age*. Firstly, the ITI-GEN model is trained using the same methodology as the original paper. Secondly, 540 images are generated in total (30 images for each combination of *age* and

---

[3]https://drive.google.com/drive/folders/1_vwgrcSq6DKm5FegICwQ9MwCA63SkRcr

*gender* category). This experiment uses the code from the original paper for both training and image generation. Evaluation is conducted using the same classifier as mentioned in the original paper, proposed by Karkkainen & Joo (2021), and is publicly accessible on GitHub[4]. To ensure a fair comparison, we established our own baseline for Vanilla SD.

**Experiment - Claim 4**   To address the quality of the generated images, we examine the FID score. It is important to note that a substantial sample size, typically exceeding 50K, is crucial to avoid overestimation of this score (Chong & Forsyth, 2020). Moreover, Stein et al. (2023) found that human judgement of perceptual realism in diffusion models does not align with the FID metric. Given the computational demands and divergence from human evaluation, leveraging the FID score poses challenges. For reproducibility, we explore the feasibility of calculating FID scores in a low-resource setting. Additionally, we assess FID score differences for positive and negative instances of an attribute, aiming for minimal divergence to ensure inclusivity.

We compute FID scores for 50, 100 and 400 vanilla SD images, generated with the prompt "a headshot of a person", to examine the impact of image quantity. Furthermore, we investigate FID scores for 100 ITI-GEN generated images per category.

**Additional Experiment - Inclusive prompt embeddings**   The authors argue that instead of specifying attributes explicitly using descriptive natural language, images can represent specific concepts or attributes more efficiently. Consequently, inclusive prompt embeddings are acquired using images as guidance. Therefore, we want to examine the distinction between utilising image and text guidance. To achieve this, we investigate the variance between inclusive prompt (image guided) embeddings and the hard prompt (text guided) embeddings.

This experiment involves three binary attributes. For each category, both the trained ITI-GEN and the hard prompt CLIP embeddings are extracted. Subsequently, these embeddings are projected onto the CLIP embedding space. To create a 2D visualisation of the projected embeddings, we employ the dimensionality reduction technique UMAP (McInnes et al., 2020).

**Additional Experiment - The inclusivity of untargeted attributes**   ITI-GEN provides inclusivity for a targeted attribute (inclusive prompt) while overlooking inclusivity for all other attributes. It would be undesirable if introducing a targeted attribute resulted in reduced inclusivity for the untargeted attributes. Additionally, the authors acknowledge a potential limitation, emphasising the risk of biases introduced by a limited set of reference images used in training ITI-GEN. Therefore, we examined the inclusivity of the untargeted attributes.

We employed HPS and ITI-GEN to create images focusing on the targeted attributes: *gender* and *age*. Subsequently, these images were classified via CLIP according to an untargeted attribute, assessing how targeting attributes might affect the distribution of untargeted attributes. For example, we generated images for the category *male* with HPS and ITI-GEN and examined how many of those images resulted in *pale* or *dark* skin. This methodology enables an analysis of how such targeted adjustments influence various unanticipated aspects within the images.

**Additional Experiment - Influence of reference images on untargeted attribute inclusivity**   In the aforementioned experiment, we investigate the inclusivity of untargeted attributes. This leads to a subsequent research question: do the observed biases stem from the selection of reference images? Consequently, we investigate whether the choice of reference images is the root cause of biases. We hypothesise that biases within the reference images are amplified in the generated images with ITI-GEN embeddings, particularly for attributes with a clear visual distinction (e.g. *gender*).

The experiment focuses on the targeted attribute *eyeglasses* and the untargeted attribute *gender*. First, the reference images of this attribute are visualised in the CLIP embedding space. Next, we construct a new set of 25 reference images, ensuring balance across both *eyeglasses* and *gender* (see Appendix E for more

---

[4]https://github.com/dchen236/FairFace

details). Afterwards, we train the ITI-GEN model on both the balanced and unbalanced reference sets and generate images for each setting. Finally, we computed the ratio for the untargeted *gender* attribute for these images.

### 3.6 Computational requirements

All experiments were conducted using a NVIDIA A100 GPU and a NVIDIA T4 GPU. Training ITI-GEN with a single attribute takes approximately 5 minutes.

## 4 Results

### 4.1 Reproducibility study

Table 1: **Comparison of ITI-GEN with baseline methods with (a) single attribute and (b) multiple attributes.**

| Method | (a) Single Attribute | | | | | | (b) Multiple Attributes | | |
|---|---|---|---|---|---|---|---|---|---|
| | $\mathbb{D}_{\mathrm{KL}}^{\mathrm{male}} \downarrow$ | $\mathbb{D}_{\mathrm{KL}}^{\mathrm{young}} \downarrow$ | $\mathbb{D}_{\mathrm{KL}}^{\mathrm{pale\ skin}} \downarrow$ | $\mathbb{D}_{\mathrm{KL}}^{\mathrm{eyeglass}} \downarrow$ | $\mathbb{D}_{\mathrm{KL}}^{\mathrm{mustache}} \downarrow$ | $\mathbb{D}_{\mathrm{KL}}^{\mathrm{smile}} \downarrow$ | $\mathbb{D}_{\mathrm{KL}}^{\mathrm{male \times young}} \downarrow$ | $\mathbb{D}_{\mathrm{KL}}^{\mathrm{male \times young \times eyeglass}} \downarrow$ | $\mathbb{D}_{\mathrm{KL}}^{\mathrm{male \times young \times eyeglass \times smile}} \downarrow$ |
| SD 5 | 0.264 | 0.615 | 0.201 | 0.136 | 0.101 | 0.069 | 0.834 | 1.023 | 1.125 |
| SD 1 | $5 \times 10^{-5}$ | 0.637 | 0.110 | 0.541 | 0.525 | 0.288 | 0.595 | 1.177 | 1.506 |
| HPS | 0.000 | 0.000 | $2 \times 10^{-4}$ | 0.693 | 0.637 | 0.018 | 0.001 | 0.098 | 0.277 |
| **ITI-GEN** | $5 \times 10^{-5}$ | $8 \times 10^{-4}$ | 0.000 | 0.0018 | 0.316 | 0.000 | 0.001 | 0.305 | 0.337 |

**Claim 1 & 2: Inclusiveness in text-to-image generation and scalable to multiple attributes** The results presented in Table 1 indicate that ITI-GEN demonstrates near-perfect performance in balancing almost every single and multiple attribute, thereby validating the first and second claim made by the authors. Moreover, ITI-GEN exhibits comparable or superior performance to HPS, providing support for the authors' assertion that utilising images as guidance, rather than language, may prove advantageous in describing categories. The qualitative results of this experiment are shown in Appendix C.

Notably, our findings reveal a $\mathbb{D}_{\mathrm{KL}}^{\mathrm{mustache}}$ score of 0.316 for ITI-GEN, while the original paper reported a value of $4.5 \times 10^{-4}$. One plausible explanation for this discrepancy is that the authors might have employed an evaluation technique different from CLIP, as indicated in their paper. Specifically, the issue of negative prompts (e.g. *without mustache*) in CLIP may lead to inaccurate attribute classification. However, upon manually evaluating the *mustache* attribute, two people classifying the generated images independently, 99 out of 100 contained the desired category. The $\mathbb{D}_{\mathrm{KL}}$ score obtained using this statistic, $5 \times 10^{-5}$, aligns with the one reported by the authors. This outlier does not contradict their claim; however, it does highlight a shortcoming in their evaluation technique.

Moreover, an analysis of the reproducibility of the vanilla SD baseline is provided in Appendix B.

**Claim 3: Scalable to multi-category attributes** Our results for the third claim align closely with those presented in the original paper. As illustrated in Figure 1, the generated images from ITI-GEN are more uniformly distributed across the multi-category attribute *age* than the baseline SD. Despite this alignment, the distributions from our experiment deviate significantly more from a uniform distribution than those reported in the original paper. Specifically, the proportion of the *age* category *20-29* for both SD and ITI-GEN is considerably higher than that of the other categories. This discrepancy may be attributed to the difference in the number of images generated and evaluated between our study and the original paper, as the authors do not provide the number of images employed in their experiment. The qualitative results of this experiment are shown in Appendix C.

**Claim 4: High-quality images** The quality of the images generated with single and multiple attributes is evaluated. Table 2 displays the FID scores corresponding to images generated across various ITI-GEN categories, compared with SD baselines evaluated on a varying number of images.

Initially, it is noteworthy that both the baseline and ITI-GEN images exhibit FID scores considerably higher than the reported 67.4 and 60.4 from the original paper for SD and ITI-GEN, respectively. However, the

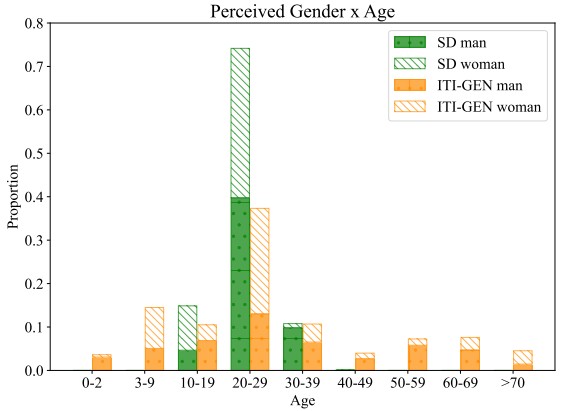
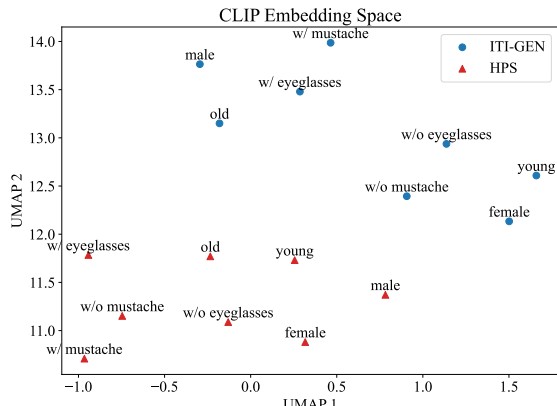

Figure 1: **Multi-category distribution** for Perceived Gender × Age with "a headshot of a person".

Figure 2: **Prompt embedding visualisation.** Two-dimensional CLIP embedding space that visualises the hard prompt and ITI-GEN embeddings.

Table 2: **FID (↓) comparison.** FID scores for the SD baseline are computed using the indicated number of images, while ITI-GEN FID scores are determined based on 100 images per category.

| SD | | | ITI-GEN | | | | | | | | | | | |
|---|---|---|---|---|---|---|---|---|---|---|---|---|
| # images | | | Male | | Young | | Eyeglasses | | Male x Young | | | |
| 50 | 100 | 400 | + | - | + | - | + | - | ++ | + - | - + | - - |
| 165.9 | 146.3 | 109.6 | 165.7 | 159.5 | 192.4 | 174.7 | 174.9 | 179.1 | 161.0 | 192.7 | 175.5 | 149.8 |

FID score is dependent on the number of images (section 3.5). This is additionally endorsed by the results obtained for SD, where a substantial decline in the FID score is evident with an increase in the number of images. Hence, considering the assumption that the authors employed 40K images for score calculation compared to our 100, the substantial deviation in FID scores can be ascribed to its dependence on quantity.

Ideally, a uniform distribution should not only be maintained in the number of images per category, but also in the quality of generated images across different categories. Upon examination of Table 2, this uniformity appears to be lacking. Particularly, there is a considerable disparity in image quality between *young positive* and *negative* categories, which is deemed undesirable.

## 4.2 Results beyond original paper

**Inclusive prompt embeddings** Examining Figure 2 reveals several observations. Firstly, it is visually apparent that the embeddings associated with the HPS distinctly differ from those of ITI-GEN. The clear visual separation in the figure underscores the unique characteristics exhibited by the HPS embeddings compared to their ITI-GEN counterparts. Secondly, a notable distinction in the treatment of category absence is evident. Specifically, the HPS embeddings of *w/ eyeglasses* and *w/o eyeglasses* are proximal, implying a similarity between the vectors. In contrast, the ITI-GEN embeddings of the same categories are more distant in the embedding space, suggesting a more accurate representation of distinct features. These results offer confirmation to the authors' claim that relying on images for guidance, as opposed to language, may be beneficial in describing categories. Lastly, the figure illustrates that, in the case of ITI-GEN embeddings, specific clusters are more well-defined compared to HPS. For instance, the embeddings of *old*, *w/ mustache*, *male*, and *w/ eyeglasses* form a cohesive group within the space. This observation raises concerns about potential introduction of unintended biases within the ITI-GEN vectors. This issue is further researched in the following paragraph.

**The inclusivity of untargeted attributes** In this experiment we investigated the inclusivity of untargeted attributes. Figure 3 highlights some insightful results.

In Figure 3a, targeting on *gender*, the impact of ITI-GEN on untargeted attributes is evident for *men*, showing increased inclusivity across all attributes. However, for *women*, this improvement of the *age* attribute is less pronounced and the *smile* attribute does not improve.

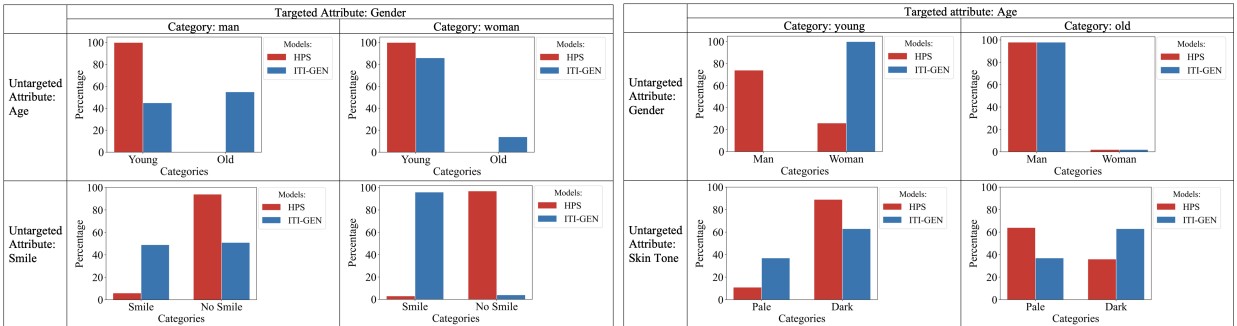

(a) Proportions of *Age* and *Smile* in HPS and ITI-GEN for the Targeted Attribute *Gender*.

(b) Proportions of *Gender* and *Skin Tone* in HPS and ITI-GEN for the Targeted Attribute *Age*.

Figure 3: **The inclusivity of untargeted attributes.** Comparative analysis of the proportion of untargeted attributes of images generated with HPS and ITI-GEN. A proportion nearing 50% signifies a balanced level of inclusivity.

As shown in Figure 3b, the results are less straightforward when moving to the *age* attribute. Within the *young* category, ITI-GEN exclusively generates *women*, leading to a reduction in inclusivity. However, there is a distinct improvement in inclusivity observed for the *skin tone* attribute. In the *old* category, inclusivity does not improve for this attribute.

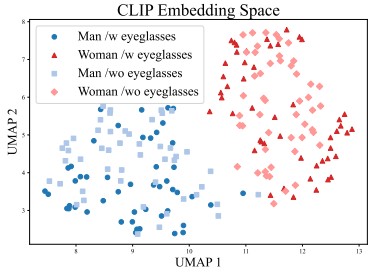

Figure 4: **Reference image embeddings.** Reference image set of *eyeglasses*, visualised in the two-dimensional CLIP embedding space.

Figure 5: **The inclusivity of untargeted attributes for balanced and unbalanced reference image sets.** Comparative analysis of the proportion of the untargeted attribute *gender* of images generated with HPS, unbalanced ITI-GEN and balanced ITI-GEN.

**Influence of reference images on untargeted attribute inclusivity** To research a potential cause of the aforementioned bias introduction, we examined the influence of the reference image set on the inclusivity of untargeted attributes.

From Figure 4 it becomes evident that within the *eyeglasses* attribute there is a clear distinction between men and women. Particularly, this demarcation is considerably more pronounced than the seperation based on the presence or absence of eyeglasses. This observation suggests that the *gender* attribute exhibits superior visual distinguishability compared to the *eyeglasses* attribute, making it a visually stronger attribute.

Figure 5 compares the *gender* distribution in images generated by ITI-GEN when trained on an unbalanced versus a balanced reference set for the target attribute *eyeglasses*. It is observed that in the case of an unbalanced dataset, all generated images targeting *eyeglasses* depict *men*, while those *without eyeglasses* predominantly portray *women*. However, when the model is trained on a balanced reference set, this bias is hardly present. This suggests that when targeting on subtle attributes, the bias of visually stronger untargeted attributes within the reference set is amplified during the image generation process.

# 5 Discussion

Throughout this work, we have conducted several experiments to reproduce the main results from the research by Zhang et al. (2023). The results provide support for their claims, as we were largely able to reproduce the original results. Specifically, our results showed that ITI-GEN produces images with significantly lower distribution discrepancies compared to baseline methods for binary, multiple and multi-category attributes.

However, some minor discrepancies emerged. Particularly, the $\mathbb{D}_{\mathrm{KL}}^{\mathrm{mustache}}$ exhibited a notably higher score compared to the original paper. This divergence may be attributed to the utilisation of a different evaluation technique. Additionally, the distributions supporting claim 3 deviate from a uniform distribution as reported in the original paper. The number of images utilised in this experiment might explain this difference, as this detail was missing in the original paper.

Upon investigating the image quality, we found the FID score to be dependent on the number of images, supporting Chong & Forsyth (2020). Additionally, the FID score demonstrated unequal results within attributes, which is undesirable. It is crucial to acknowledge that due to constraints in our computational resources combined with the FID score's quantity dependence, definitive conclusions cannot be drawn from the reported scores. Nevertheless, we strongly encourage future researchers to consider the objective of achieving uniform image quality across categories.

Interestingly, comparing text- and image-guided embeddings supported the authors' motivation that utilising image guidance can be beneficial in category description. However, it concurrently raised concerns regarding the potential introduction of additional biases.

The investigation of the inclusivity of untargeted attributes showed increased inclusivity for some and decreased for other attributes compared to HPS. Especially, the decrease raises concern. This shows that ITI-GEN only guarantees inclusivity for the specified targeted attributes. The fact that ITI-GEN is trained on a limited set of reference images potentially explains these results due to the limited possibility of diversity in the training data.

Further investigation into the aforementioned influence of the reference image set revealed that the presence of bias in visually strong attributes within this set can give rise to additional biases in the generated images. Notably, training ITI-GEN with a balanced reference image set resulted in a decrease in additional biases. This demonstrates the importance of a balanced reference image set. In conclusion, it is crucial to remain mindful of potential additional biases that might be introduced by the reference images.

Overall, the experiments from the original paper were largely reproducible, and their main claims proved to be valid. Our research showed that aiming for an inclusive reference set could help to achieve the goal of inclusiveness among all attributes.

## 5.1 What was easy, and what was difficult.

The original code and dataset were publicly accessible, well-organised and thoroughly documented. Additionally, detailed flags, encompassing all hyperparameters, were provided. The original appendix contained an extensive array of supplementary results, addressing nearly every outstanding concern. Notably, the low number of parameters in the ITI-GEN model rendered it highly efficient for training. Consequently, replicating the principal findings proved to be straightforward.

The primary challenge encountered was the computational demands associated with image generation using the diffusion model. Unfortunately, we were therefore unable to generate as many images as the original work did for reproducing the scores. Generally, this limitation did not pose a significant obstacle in validating their primary assertions, as the results exhibited a consistent trend for the KL divergence metric. However, in instances involving the FID score, which is particularly sensitive to the quantity of images, we were unable to replicate the author's results due to the constrained number of generated images.

## 5.2 Communication with original authors

We contacted the original authors to ask about the apparent disparity in generated image quality. Their prompt response included comprehensive details regarding the process of image generation.

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

## A Overview of image generation using HPS and ITI-GEN

In Figure 6 a general overview of image generation using HPS and ITI-GEN is shown, highlighting their differences.

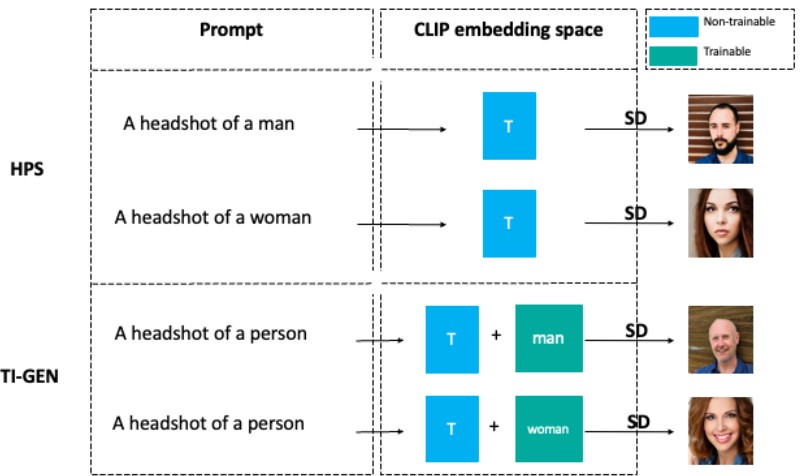

Figure 6: **A general overview of image generation using HPS and ITI-GEN**

## B Vanilla SD reproducibility

In our attempt to reproduce the outcomes of the baseline as described in the original paper, we observed a deviation in the results from those reported in the original study. While there is a discrepancy in the precise quantitative scores when compared to the original paper, the general trend observed in the scores largely aligns with the original findings. However, this conformity does not extend to the scores of the specific attributes, notably *eyeglasses* and *smiling*, where we noted a significant divergence. This divergence in scores could potentially be attributed to variances in the computational resources utilised in our replication study and the number of images used to calculate the $\mathbb{D}_{\mathrm{KL}}$ score, as the authors did not provide precise details on this matter.

## C Qualitative results

In Figure 7, a distinct separation is evident among the four defined categories. However, when including additional attributes, the distinction between the categories becomes somewhat ambiguous, as illustrated in Figure 8. For instance, the attributes *eyeglasses* and *perceived gender* are not completely uniformly distributed. A better separation between categories occurs when expanding the analysis to multi-category attributes, as depicted in Figure 9. The distinction between men and women remains prominent, accompanied by a visible increase in age.

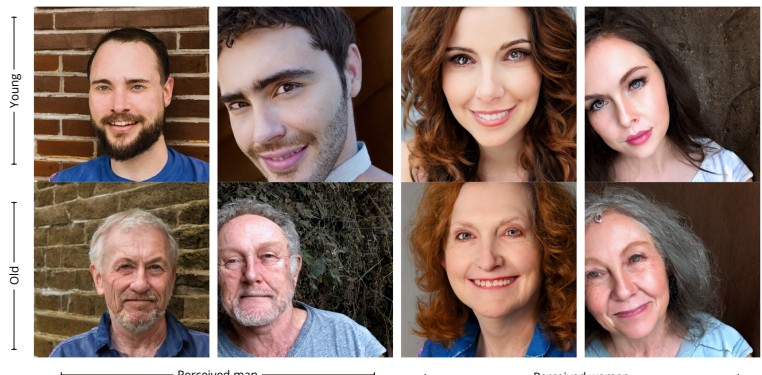

Figure 7: **Results of ITI-GEN on binary attributes** for Perceived Gender × Age. Examples are randomly picked with "a headshot of a person".

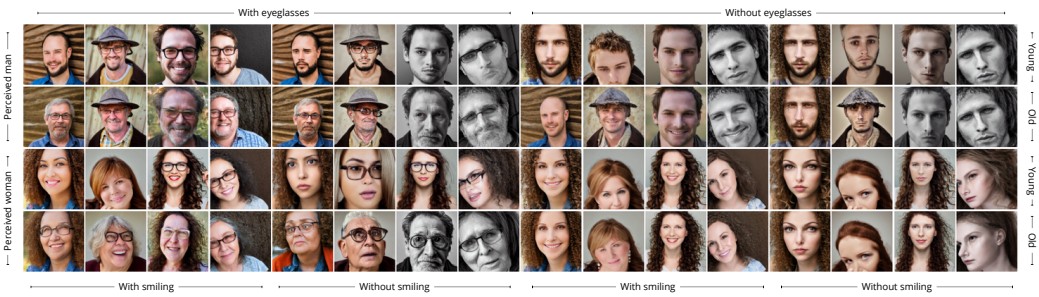

Figure 8: **Results of ITI-GEN on the combination of four binary attributes** for Perceived Gender × Age × Eyeglasses × Smiling. Examples are randomly picked with "a headshot of a person".

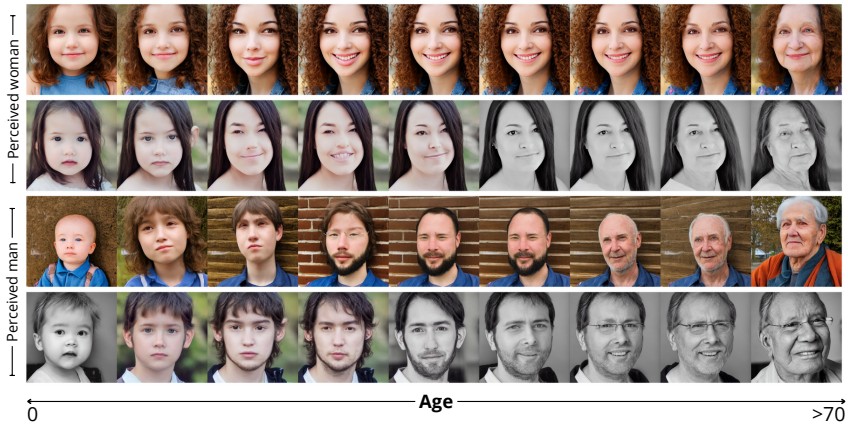

Figure 9: **Results of ITI-GEN on multi-category attributes** for Perceived Gender × Age. Examples are randomly picked with "a headshot of a person".

Table 3: **Hyperparameters** configuration for training the ITI-GEN model.

| | learning rate | $\lambda$ | steps per epoch | learnable tokens | epochs | batch size |
|---|---|---|---|---|---|---|
| Train | 0.01 (0.005 after 10 epoch) | 0.8 | 5 | 3 | 30 | 16 |

## D  Hyperparameters

The ITI-GEN model's training is fine-tuned with hyperparameters: an initial learning rate of 0.01, halved after 10 epochs for precision adjustments, and a lambda of 0.8. The training spans 30 epochs, with updates every 5 steps, targeting 3 specific learnable tokens. A batch size of 16 ensures a balance between computational speed and memory efficiency.

For image generation, the only hyperparameters are the DDIM steps, set to a default of 50, and the batch size, which was limited to 1 due to memory constraints.

## E  Balancing the reference image set

Table 4: **Reference image set distribution.** This table presents the composition of the image sets of the attribute *eyeglasses* classified by CLIP based on *gender*. The balanced image set is a subset of the reference image set.

| | Eyeglasses + | | Eyeglasses - | |
|---|---|---|---|---|
| | Female | Male | Female | Male |
| Reference image set | 53 | 147 | 114 | 86 |
| Balanced image set | 13 | 12 | 13 | 12 |

