# OpenReview forum: "Reproducibility of "ITI-GEN: Inclusive Text-to-Image Generation""
_TMLR — Rejected by TMLR_

### Review · Reviewer_reRT · 2024-03-23

**Summary Of Contributions:**

This paper replicates the "ITI-GEN: Inclusive Text-to-Image Generation" using the officially released code. The experiments validate the original claims and reproduce the results. Additionally, several ablation studies are conducted to enhance understanding of the original algorithm.

**Audience:**

No

**Broader Impact Concerns:**

Not applicable.

**Claims And Evidence:**

Yes

**Requested Changes:**

As outlined in the TMLR Acceptance Criteria (https://jmlr.org/tmlr/acceptance-criteria.html), a thorough reproducibility report that methodically investigates the robustness or generalizability of ITI-GEN is required.

**Strengths And Weaknesses:**

Pros:

This paper validates the proposed claims in ITI-GEN, which is beneficial for researchers seeking to validate their own claims.

Cons:

Given that the official code is publicly accessible and no significant new experiments were performed for an in-depth analysis of ITI-GEN, this reproducibility paper appears more akin to a student project rather than a research paper. Furthermore, the limitations of ITI-GEN introduced in this paper were already acknowledged in the original publication.

---

### Review · Reviewer_nEzy · 2024-03-28

**Summary Of Contributions:**

The paper looks into the reproducibility of ITI-GEN as an attempt to validate the main claims presented by the authors of ITI-GEN. During the process this paper highlights the importance of a balanced dataset of untargeted attributes for embedding training, as those biases are picked up by the embeddings

**Audience:**

Yes

**Claims And Evidence:**

Yes

**Requested Changes:**

1. Use precision and recall to disentangle image quality from diversity
2. As the embeddings learned are solely from CLIP, it would be good to show results on different stable diffusion variants that train with clip embeddings to show that these learned embeddings are generalizable to any diffusion model
3. While an unbalanced dataset affects the inclusiveness of untargeted attributes, experiments should be done where the learned embeddings for those untargeted attributes are trained and used to see if such a naive approach effectively removes the problem, thereby not needing a balanced dataset with regard to untargeted attributes but simply adding new attribute tokens when such inclusivity is wanted.

**Strengths And Weaknesses:**

**Strengths:**

1. Bringing light to the different values in the 'mustache' attribute case(Table 1) is interesting. Could more clarification on this be given?
2. The paper additionally highlights the difference between the learned embeddings of HPS over ITI-GEN.
3. Additionally, this paper also highlights the importance of a balanced dataset of untargeted attributes for embedding training, as those biases are picked up by the embeddings.

**Weakness:**

1. Images used for FID calculation are low, resulting in high values that might be inconclusive
2. For image quality assessment of ITI-Gen, it might be better to report precision and recall (https://arxiv.org/abs/1904.06991) instead of FID as it disentangles image quality and image diversity in image generation to better understand how it affects the generation of diverse samples
3. While the difference between the HPS and ITI-GEN learned embeddings is interesting, more analysis should be done to see how they behave differently.

---

### Review · Reviewer_ppcM · 2024-04-30

**Summary Of Contributions:**

This submission focuses on reproducibility of the paper 'ITI-GEN: Inclusive Text-to-Image Generation' analyzing the claims by the authors and evaluating them. The reproducibility study conducted supports the main claims of ITI-GEN, showing evidence of improved inclusiveness in generated images. However, additional experiments reveal a limitation of ITI-GEN, indicating that it only guarantees inclusivity for the specified targeted attributes, suggesting the need for further refinement and exploration in achieving broader inclusivity.

**Audience:**

No

**Broader Impact Concerns:**

Concern on Ethical Implications: The modifications made to the experimental setup due to limited resources, as discussed in the submission, could misrepresent the reproducibility and reliability of the ITI-GEN results. This oversight raises ethical concerns about the accuracy and transparency of the research findings, which could mislead subsequent research and applications. A Broader Impact Statement should address these potential ethical implications to ensure clarity and integrity in the reporting of research outcomes.

**Claims And Evidence:**

No

**Requested Changes:**

Adjustment of Experimental Setup to Match Original Conditions (Critical):
The authors need to address the critical issue of altering the experimental conditions, such as the reduction of DDIM steps from 50 to 25 and conducting fewer experiments than the original ITI-GEN paper due to GPU resource limitations. These changes compromise the reproducibility study's goal to accurately verify the original claims and assess the technology’s robustness. To secure a recommendation for acceptance, it is essential that the authors either access the necessary resources to replicate the original experimental conditions or justify the modifications with detailed analyses on how these changes might affect the results.
Expansion of Experimental Scope (Strengthens the Work):
Extending the experimental setup by including additional prompts and settings, as suggested for Claims 1 & 2, would strengthen the submission. This is not critical for acceptance but would enhance the thoroughness of the reproducibility study and provide deeper insights into ITI-GEN's performance under varied conditions.
Increased Sample Size in Experiments (Critical):
For Claim 4, where the FID score's accuracy is crucial, the use of a substantially larger sample size is necessary. The current use of 50, 100, and 400 samples is insufficient to draw reliable conclusions, particularly when the original study suggests a sample size exceeding 50K. Increasing the sample size is critical to avoid misrepresentation of the FID score and to ensure the validity of the study’s findings.
Transparent Reporting of Limitations (Strengthens the Work):
In Section 5, the authors acknowledge limitations due to computational resources, which impact the FID score’s reliability. It would strengthen the paper to include a more detailed discussion of how these limitations affect the results and what measures were taken to mitigate their impact. This addition would provide clarity and transparency, enhancing the credibility of the research.
Methodological Justifications for Changes (Strengthens the Work):
Where experimental conditions were altered, such as in the reduction of DDIM steps and the selection of only one challenging setting in Claim 3, providing a robust justification for these decisions is essential. This adjustment would strengthen the work by clarifying the rationale behind these changes and their expected impact on the study's outcomes.

**Strengths And Weaknesses:**

Strengths:
A successful reproducibility study of research work requires that the researchers: 1) verify the original claims of ITI-GEN, 2) conduct additional experiments to provide further evidence supporting these assertions, and 3) thoroughly demonstrate any limitations. The authors have made efforts to achieve this; however, they have introduced shortcomings in the experimental setup.

Weaknesses:
The authors chose an approach of changing the experimental setup citing a lack of computational resources, which adds significant confusion in understanding the original work ITI-GEN's contributions and drawbacks and does not support the broader goal of reproducibility.

Section 3.4, Hyperparameters: The authors mention that 'Due to limited time and resources, the Denoising Diffusion Implicit Models (DDIM) steps of SD have been decreased from 50 to 25 for the experiment of the third claim. The DDIM framework iteratively improves samples via denoising. Reducing its steps from 50 to 25 speeds up the sampling process but could slightly alter the quality.' My argument is that while reproducibility is the main objective of the paper, changing sampling steps, which can significantly alter the results, renders these results unfair.

Experiments - Claims 1 & 2: The authors claim, 'Due to limitations in GPU resources, we were unable to execute every experiment with each of the five prompts outlined in the original paper.' The goal of a reproducibility study is to verify the claims of the original submission, ITI-GEN, to uncover the contributions and shortcomings of ITI-GEN. However, the authors of the current submission changing the experimental section does not serve the overarching purpose. Instead, the authors should consider extending the original experimental setup by adding ablations to assess the impact of additional prompts, thereby covering a more comprehensive experimental section.
Experiment - Claim 3: The authors chose to 'reproduce one of the two challenging settings.'

Experimental - Claim 4: The authors mention that 'a substantial sample size, typically exceeding 50K, is crucial to avoid overestimation of this score,' referring to the FID score. However, the authors use very limited samples of 50, 100, and 400 vanilla SD images. I do not see a proper justification for the rationale here.

Section 5, Discussion: The authors mention that 'It is crucial to acknowledge that due to constraints in our computational resources combined with the FID score’s quantity dependence, definitive conclusions cannot be drawn from the reported scores.' Once again, the goal of this work should be to provide more clarity in the experimental results of the original work, ITI-GEN.

---

### Decision · Action_Editor_7C9v · 2024-06-03

**Recommendation:** Reject

**Comment:**

This paper tries to reproduce ITI-GEN, an inclusive text-to-image generation models. By some experiments, it is claimed that the claims of the original paper are mostly valid. However, the reviewers have raised a number of questions, with the most concerned one being that the evaluation settings are significantly different from the original paper, thus the conclusions are not well supported. Unfortunately there is no rebuttal for the reviews. Thus, I view all the reviewer concerns to be valid and propose rejection of the paper.

**Audience:**

Audience in text-to-image generation might find the work valuable.

**Claims And Evidence:**

The paper is a reproducibility paper that studies previous models. Unfortunately the claims are not well supported by the enough evidence. See some more details below.